# Regenerative Models for the Integration and Regeneration of Head Skeletal Tissues

**DOI:** 10.3390/ijms19123752

**Published:** 2018-11-26

**Authors:** Warren A. Vieira, Catherine D. McCusker

**Affiliations:** Department of Biology, University of Massachusetts Boston, Boston, MA 02125, USA; Warren.Vieira@umb.edu

**Keywords:** regenerative medicine, transplantation, jaw regeneration, limb regeneration, positional information, zebrafish, axolotl

## Abstract

Disease of, or trauma to, the human jaw account for thousands of reconstructive surgeries performed every year. One of the most popular and successful treatment options in this context involves the transplantation of bone tissue from a different anatomical region into the affected jaw. Although, this method has been largely successful, the integration of the new bone into the existing bone is often imperfect, and the integration of the host soft tissues with the transplanted bone can be inconsistent, resulting in impaired function. Unlike humans, several vertebrate species, including fish and amphibians, demonstrate remarkable regenerative capabilities in response to jaw injury. Therefore, with the objective of identifying biological targets to promote and engineer improved outcomes in the context of jaw reconstructive surgery, we explore, compare and contrast the natural mechanisms of endogenous jaw and limb repair and regeneration in regenerative model organisms. We focus on the role of different cell types as they contribute to the regenerating structure; how mature cells acquire plasticity in vivo; the role of positional information in pattern formation and tissue integration, and limitations to endogenous regenerative and repair mechanisms.

## 1. Introduction

The skeletal bones of the face and head play multiple roles that are essential to our survival. They encase and protect our fragile brain tissue from injury. They are essential for respiration, the intake and mastication of food, and verbal communication. They separate our eyes and ears at an appropriate distance, so that we can achieve binocular vision and resolve the spatial origin of sound, respectively. Given these diverse and essential functions of the head skeleton, traumatic injury to these tissues can have a profound impact on our health and quality of life. Although humans have limited regenerative capacity of these structures, different surgical and regenerative treatments have been developed to repair these injuries. However, the outcome of these treatments can vary greatly depending on the age of the patient and the location and size of the injury. In this review we will discuss (1) the most recent advances in regenerative technologies to replace or repair damaged cranioskeletal tissues with a focus on the jaw, (2) how endogenous regeneration of the jaw occurs in regenerative model organisms, and (3) how we can apply what is known about endogenous regeneration of the jaw and other skeletal elements in the body to develop more effective regenerative therapies.

## 2. Current Regenerative Therapies for the Jaw

Damage to the human mandibular and maxillary jaw bones can occur through a variety of mechanisms, including diseases, such as cancers, or direct trauma. Moreover, tooth loss or extraction will lead to the alveolar ridge exhibiting atrophy, both in height and width, which impedes implantation; although, the basal processes of the mandible and maxilla will remain unaffected [1]. Therefore, bone maintenance and/or regeneration under these circumstances is essential to ensure jaw functionality and allow for the restoration of dentation by implantation where necessary. 

A variety of different methods have been pursued to replace or repair damaged jaw tissue, with varying degrees of success. One method that has worked well for repairing missing jaw tissue in some patients has been through distracting the remaining mandibular bone. Distraction is the process whereby new bone is generated by applying stress (stretching) to the bone. Some positive aspects of using distraction are (1) that the new bone tissue is fully integrated and (2) there is an expansion of the other surrounding tissues, including skin, muscle, nerve, blood vessels, periosteum, and cartilage. Additionally, since this method utilizes endogenous tissue, the issues of tissue rejection can be avoided. Distractors can be internal or external and are directly fixed on the bone. The method of distraction has been used successfully on children with congenital craniofacial deformities [2], as well as reconstructing mandibular defects [3]. However, in instances where there is not enough of the mandible to distract, for example in some cancer and trauma patients, this method of treatment is not an option.

In circumstances where minimal alveolar or mandibular bone remains, many studies have tried repairing the damaged jaw tissue by using implants or tissue transplants from different anatomical locations. Some of the earlier attempts involved alloplastic implants of a structure composed of an inert metal such as steel, titanium, or chrom-colbalt steel. One of the many issues with these types of implants is that the material is not composed of cells, and so the implant is incapable of healing and fully integrating with the surrounding tissues. Not surprisingly, these implants had a high failure rate and were removed as a result. Hybrid implants consisting of metal trays with “cancellous bone” were also tested and associated with similar, poor outcomes (reviewed in [4]). 

Over the last few decades, much work has focused on the development of different synthetic bone substitutes, many of which have led to good healing outcomes (reviewed in [5]). Implants composed of these substitutes have the advantage that they alleviate the need to harvest bone from a different region of the patient’s body, and can be “tunable” to the needs of a particular injury site. For example, the porocity and topography [6] of these scaffolds can be easily manipulated, biologically active molecules such as specific extracellular matrix molecules or growth factors [7,8,9,10] can be added, and specific cell types such as osteoblasts [11], mesencymal stem cells [12] and endothelial cells [13] can be seeded to generate patient and need-specific implants. 

Osteomyocutaneous flaps, which are composed of vascularized muscle (ex. trapezius or latissimus muscle) that sometimes includes a portion of the underlying bone, have been used for head, neck, and jaw defects. These transplantations have been moderately successful in repairing the damaged tissue; however, patients often suffer from complications associated with the removal of the donor tissue. Vascularized bone flaps, also known as free flaps, where the graft tissues have been harvested from a variety of locations including ribs, iliac crest, fibula, or metatarsals have also been used. Early trials resulted in high morbidity, surgical complications, and a total or partial bone loss of the graft. More recent trials with the fibula with fixed plates to the mandible have been relatively successful, with remarkable integration reported between the tissue transplant and the mandibular defect [14,15]. Similar results have been reported from iliac crest and scapular spine implants [16].

Currently, the most frequently preformed methods for jaw repair involve autologous bone grafts; and is associated with 100–76.4% success and 100–86.9% survival rates [17,18]. Common procedures in this context involves either the direct implantation of the donor bone, or grinding thereof, and subsequently seeding the resultant particulate matter into a scaffold, prior to implantation (also referred to as bone graft substitutes). The latter procedure is beneficial when donor tissue is limited and would require remodel into an appropriate shape for implantation.

Multiple aspects appear to contribute to the success of autologous bone transplantation in the context of jaw repair. Donor tissue has been derived from intraoral (maxilla, mandible) or more distal (iliac crest, pelvic or calvarial) sites [19]; however, they are not equivalent in terms of their restorative capacities. Calvarial bone grafts, for example, show better stability after oral implantation relative to iliac bone, but graft segments are thinner and exhibits worse vertical bone growth [20]. Jaw-derived grafts are associated with better graft survival, osteogenic activity and less volume loss relative to iliac bone [19]; and better mineralization, albeit less than the original jaw tissue, compared to pelvic grafts [21]. These donor site differences in regenerative capabilities may, in part, be due to the fact that mammalian bone mesenchymal stem cells exhibit skeletal site-specific variations in terms their osteogenic potential [22,23]. In humans and rodents, maxilla and mandible bone derived mesenchymal stem cells (MSCs) exhibit greater proliferative and survival properties, and significantly higher in vitro and in vivo osteogenic capacity, relative to iliac crest and tibia derived counterparts [22,23]. In rodents, direct implantation of mandible derived MSCs, seeded onto a scaffold, into a mandible defect resulted in more new bone formation relative to tibia derived MSCs or scaffold-alone grafts [23].

In the case of alveolar bone regeneration, local augmentation by guided bone regeneration is commonly employed with success. This has been extensively reviewed elsewhere [24,25] but briefly, this surgical protocol involves the implantation of a resorbable membrane into the defected area which mechanically allows for the invasion of osteogenic but no other competing non-osteogenic tissues. Although non-resorbable membranes show poor soft tissue compatibility, their use is associated with a reduction in new bone resorption and the generation of harder, stiffer and more mineralized osteoblast-derived matrices relative to its resorbable counterpart. Irrespective of the degradability of the membrane, growth factors, bone, or bone-derived materials are frequently used in conjunction with this technique [24,25]. Growth factors, such as bone morphogenetic proteins and platelet-rich plasma, aid regeneration by recruiting and stimulating proliferation of osteogenic tissue [24]. The use of bone derived material in this context has varying outcomes. For example, cortical bone grafts exhibit far more resorption relative to cancellous block grafts; with the latter also exhibiting superior revascularization and strengthening during the healing process as well [25].

There is also evidence that transplantations may stimulate an endogenous regenerative response in younger patients. Striking regeneration, where a *de novo* jaw joint is formed, has been reported in an adolescent patient that was treated with a costochondral graft into a mandibular resection [26]. Still, although the outcome at treatment site was improved, patients need to recover from the removal of the harvested tissue and any complications that are involved with the loss of that tissue at the site of harvest.

More recently, researchers have attempted stem cell-based therapies. Stem cells, and differentiated osteoblasts thereof, are valuable sources of donor tissue, as it negates the risk of donor site morbidity which have been associated with bone graft harvesting [27,28,29]. Bioabsorable poly (l-lactide) (PLLA) mesh scaffolds with bone marrow have been tested for jawbone regeneration. This method had an 84% success rate but does not work well with post radio-therapy patients or elderly people who have diminished bone precursor cells [30]. A combination of buccal fat pad stem cells (from the cheek) with iliac bone grafts has been used to treat maxillomandibular atrophy, and a resultant increased bone width was reported in the experimental group [30,31]. Similarly, allogenic chord stem cells or cranial derived bone marrow stem cells combined with alloplastic graft material for mandibular reconstruction has demonstrated some bone regeneration [22,23]. Dental pulp stem/progenitor cells can be employed for jaw restoration as well. When seeded into a collagen sponge scaffold and implanted into mandible, patients exhibited vertical repair of alveolar bone and restoration of periodontal tissue [30]. Some of the major challenges of these treatments are the formation of structures with the appropriate bone pattern (without ectopic bone formation), and the integration of the newly regenerated bone with the existing bone and the surrounding soft tissues.

In summary, although both reconstructive and regenerative methods have been developed with moderate success, not all of these methods will work for all patients. Future therapies will need to focus on either improving the integration of tissue transplantations, or the stimulation of an endogenous regenerative response for better patient outcomes. In this review, we will explore the mechanisms by which regenerative model organisms perfectly integrate transplanted tissue or endogenously regenerate missing jaw tissue to identify potential targets that can be explored for regenerative therapies for humans. 

## 3. The Patterning and Development of the Cranial-Facial Skeleton

Many similarities exist between how the head skeleton is formed during embryogenesis, and how these structures can reform during regeneration. Thus, we will first discuss how the cranial-facial skeleton develops so that comparisons can be more easily made with the regenerative mechanisms.

The skeletal structures in the cranium are derived from a tissue in the developing embryo known as the cranial neural crest (CNC). The CNC is a transient population of multipotent cells that is located lateral to the neural plate on the dorsal side of the anterior end of all vertebrate embryos. During neurulation, the CNC cells delaminate from the neural folds and migrate ventrally between the ectodermal and endodermal germ layers. Depending on where the CNC cells are located on the A/P axis in the developing head they will express different positional markers, migrate on discreet pathways, and differentiate into different skeletal elements. 

During early development the hindbrain is segmented into anterior/posterior compartments known as rhombomeres, and CNC migration is directed by these underlying compartments (as reviewed in [32]). The CNC migrates in a ventral direction along three major pathways, the mandibular, the hyoid, and the proximal pharyngeal (branchial) arches. The mandibular arch migrates around the optic vesicle, will fuse at the region below the optic vesicle, and will generate most of the cartilage and membranous bone of the face and jaw. The CNC cells from the hyoid arch will migrate ventrally and differentiate into a small number of skeletal elements in the middle ear and the hyoid bone. The most posterior CNC segments migrate to populate the pharyngeal pouches, and differentiate into the “gill” and pharyngeal cartilages in the neck skeleton. 

Each arch of the CNC expresses specific patterning genes, and the expression of these genes plays an important role in the formation of specific skeletal elements in the head. The mandibular (neurocranial) arch arises from the Otx-expressing region of the hindbrain, and does not express Homeobox (*Hox*) genes. As the cells that form this arch migrate, they express distal-less homeobox genes (*Dlx*) *Dlx1/2* and *Dlx1/2/5/6* in a dorsal ventral gradient and further subdivide into the maxillary and mandibular processes, respectively [33,34,35]. The hyoid and branchial streams arise from the *Hox-2* and *Hox-3* expressing regions of the hindbrain, respectively. The alteration of the expression of these genes results in homeotic transformations. For example, the disruption of *Hoxa-2* expression in the rostral branchial (hyoid) region of the head in developing mice results in the loss of hyoid cartilages and the formation of duplicated mandibular arch-derived skeletal structures [36,37]. The reverse manipulation of overexpressing *Hox-2* in the normally Hox-negative mandibular arch cells results in the loss of jaw elements and duplication of hyoid skeletal elements [38,39].

Interestingly, grafts of cells from the mandibular (*Hox* negative) to the hyoid (*Hox* positive) arch in avian species results in the formation of ectopic skeletal tissue from the mandibular segment in the host location [40]. Thus, the lack of *Hox* expression in these cells does not preclude their identity as the mandibular segment. However, a community effect also plays an important role in determining whether or not these cells maintain their original, arch-specific identities. Transplantation of either single cells or small groups of cells from one arch to another during an early stage of CNC development in zebrafish results in the transformation of *Hox* expression from the original to the host location, and the contribution of the grafted cells to the host structures [41]. This plasticity is, however, dependent on how differentiated the cells are. Performing the same grafting manipulations in later staged embryos, where the CNC cells have differentiated, results in the grafted population maintaining their original *Hox* expression profile and facilitating the development of structures consistent with the original location from which the graft was derived [41]. Similar results have also been reported in the mouse model system [42].

Once the CNC cells reach their destination in the embryo and cease migrating they will differentiate into their fated tissue identities. Evidence indicates that the cells that are fated to become skeletal tissue provide positional cues to the surrounding cell types. For example, grafts of the mandibular segment to the hyoid segment in developing avian embryos doesn’t just result in the formation of ectopic mandibular skeleton in the hyoid segment, but also reorganizes the surrounding mesoderm to pattern and differentiate into the mandibular musculature, which integrates with the ectopic skeletal tissue [40]. The transplanted CNC cells also provide instructional cues to other non-neural crest populations including those that generate the avian beak, feathers, and tendons [43]. However, it is also clear that the surrounding epithelium is providing patterning cues to the *Hox* negative mandibular arch. Removal of the endoderm close to this arch results in the loss of cranial skeletal elements, and the addition of grafted endoderm results in the formation of ectopic cranial skeletal elements [44,45]. Thus, and interplay of patterning cues between the epithelial cells and the CNC are required for the development of this tissue.

## 4. Differentiation of the Head Skeleton

As opposed to the more posteriorly located trunk neural crest, the cranial neural crest cells contribute to skeletal tissues of the head. This ability appears to depend, at least in part, on their low *Hox* expression status [46,47]. The most caudal (anterior) CNC arch is *Hox* negative and will form most of the skeletal elements in the head and face. The hyoid arch expresses *Hox2* genes, and will only form the small hyoid and middle ear skeletal elements. Last, the posterior pharyngeal arches express both group 2 and 3 *Hox* genes, and has only a small contribution to the neck skeleton. Loss of *Hox* expression can force trunk neural crest cells to differentiate into skeleton [48].

The mechanism of osteogenesis depends on the cranium skeletal element considered (as reviewed in [49]). The flat bones of the skull that form the top and sides of the brain case, as well as the facial bones, form through a process of intramembranous ossification, whereby mesenchymal cells are converted directly into bone. The bones that compose the floor of the brain case form via endochondrial ossification, where a cartilage intermediate forms first, and acts as a scaffold for the osteogenic precursors.

Multiple molecular signaling events play an important role in the differentiation of the head skeleton. Endothelin-1 (Et-1) secretion from the epithelium and mesenchyme surrounding the pharyngeal arch, instructs the formation of dorsal/ventral skeletal elements in the head [50,51,52]. Loss of function studies in mice and zebrafish indicate that the concentration of Et-1 dictates whether a dorsal or ventral skeletal element is induced; high concentrations of Et-1 result in the formation of ventral structures, whereas low concentrations result in the formation of dorsal skeletal elements [50,53,54]. *FGF* expression is linked to CNC cell survival, migration, and also activates the expression of genes that promote CNC-specific cell differentiation, including the skeleton [55]. Additionally, high concentrations of FGF promote cartilage and bone differentiation, and can result in the formation of ectopic cartilage elements. In summary, the interplay of signaling events between the different tissues in the developing head plays an essential role in the differentiation of the craniofacial skeleton.

## 5. Vertebrate Model Systems to Study for Jaw Regeneration

While the embryonic mechanisms underlying the development of the craniofacial skeleton are mostly conserved, regenerative capabilities are highly variable amongst different vertebrate species; fish and amphibians exhibit extensive regenerative abilities of complex tissues and structures, whereas mammals and birds exhibit limited, minimal responses. As reviewed elsewhere, zebrafish [56] are commonly employed as model systems for both cardiac and neurological regeneration, while Urodele amphibians (newts and salamanders) are typically used to study limb [57] as well neurological regeneration [58]. The phenomenon of jaw regeneration has also been documented in these model organisms; however, it is less extensively studied. Interestingly, this specific phenomenon is absent in most anuran species, such as the adult African Clawed frog [59], but appears to present in at least some reptile species, such as the Marsh crocodile [60].

To facilitate better comparisons between these model organisms we will provide a brief description of the basic anatomical structure of the zebrafish and Urodele jaw. We will then discuss current findings (Table 1), interpretations, and limitations of jaw regeneration associated with the use of these different model systems. Focus will be placed on lower jaw regeneration, which has been more extensively studied relative to the upper jaw; however, work by Ghosh 1994 has shown that upper and lower jaw regeneration occur by identical process in the adult newt [61].

## 6. The Anatomy of the Lower Jaw in Regenerating Species

The lower jaws of adult zebrafish and Urodele amphibian are anatomically similar and comprises of the mandible, hyoid apparatus, brachial arches as well as the associated mandibular muscles and connective tissues. These structures are similar to those of the mammalian counterpart; however, distinct differences do exist. 

In the zebrafish and amphibian, two dermal bones, pre-articular (present on the lingual side only) and dentary, enclose a cartilaginous rod, known as Meckel’s cartilage, to form the mandible. Dorsally and ventrally the mandible curves inwards, with the distal ends joined by the median symphysis, a cartilaginous structure [59,61,62,63,64,65,66,67,68]. In the adult newt, the dentary bone of the mandible, which stretches from the basal articulation to the median symphysis, possess mature, bicuspid teeth that are arranged towards the outer, dorsal margin of the bone. Immature teeth, which will develop and migrate out with time to replace the mature teeth in the event of tooth loss, are located in the inner border of this tissue [59,61,66,68,69]. The pre-articular bone serves as a site of masticatory muscle attachment [61]. Far less ossification is present in the lower jaw of the larval amphibian, with the mandible being constituted by an elongated cartilaginous rod that becomes surrounded by bone, in a proximal to distal direction, with time. Once the dentary bone has been established, monocuspid teeth will erupt [61].

Relative to the amphibian and fish, the tissue composition of the human jaw is different. The U-shaped human mandible is formed early during embryogenesis, initially constituted by hyaline cartilaginous tissue and thick perichondral fibrous mesenchyme, derived from Meckel’s cartilage [70]. As development progresses, ossification produces intramembranous bone adjacent to the Meckel’s cartilage, which forms the architecture of the mandibular body. Endochondral ossification is associated with the mandibular condyle. By 12 weeks post-fertilization, the Meckel’s cartilage has diminished in size, and only a small portion of it, located in the mandible close to the midline near the symphysis, remains unossified and forms two nodules of fibrous tissue that occupies the dorsal surface of the symphysis [71]. Human dentition is both heterodont and diphyodont; with the eruption of the deciduous teeth being delayed till after birth and subsequently replaced, at a later time point, by permanent dentation (as reviewed by the authors of [72]).

## 7. Endogenous Regeneration of the Jaw

Regenerative mechanisms can take different forms depending on the organism and the organ or tissue that is being regenerated (as reviewed in Lismaa [73]). These mechanisms are generally divided into four categories, epimorphosis, morphallaxis, stem cell mediated, and compensatory regeneration. Compensatory regeneration occurs when differentiated cells divide but maintain their differentiated function to replace missing or damaged tissue. Human liver regeneration is an example of a regenerative process that heavily utilizes this mechanism. In stem cell mediated regeneration, stem cells generate the “building blocks” of the missing structures. The re-growth of hair shafts, from follicular stem cells, and the replacement of blood cells, from hematopoietic stem cells, are examples of this regenerative mechanism. Epimorphosis and morphallaxis are regenerative mechanisms that are generally reserved for the replacement of more complex structures. Morphallaxis occurs when the remaining undamaged tissue reorganizes into the pattern of the missing structure(s). This type of regeneration has been well documented in hyrax species that utilize morphallaxis to regenerate complete heads. Last, epimorphic regeneration occurs when mature cells dedifferentiate into progenitor cells known as blastema cells, which divide, re-pattern, and re-differentiate into the missing structure(s). Amphibian limb regeneration is an example of epimorphic regeneration. Many regenerative processes do not exclusively use one regenerative mechanism, but often use a combination of mechanisms.

## 8. Jaw Blastema Formation

Jaw regeneration in appears to occur mostly by epimorphosis, where the surrounding tissue dedifferentiates and contributes to the formation of the regenerative organ known as the blastema. Distal transverse amputation of the lower jaw in zebrafish as well as Urodele species including the Eastern Newt, the Japanese fire belly newt, the Northern dusky salamander, the Red-backed salamander, the Northern two-lined salamander, the Marbled salamander, and the Tiger salamander (*Nofophthalmus viridescens*, *Triturus viridescens*, *Cynops pyrrhogaster*, *Desmognathus juscus*, *Plethodon cinereus*, *Eurycea bislineata*; *Ambystoma opacum*, *Ambystoma Tigrinum*) induces a regenerative response (Figure 1). The initial response to this type of injury appears to be conserved, not only between species but also between body-structures (Figure 2). Prior to blastema formation, soft tissue [61,62,74], including muscle [59,64]) and dental lamina [74], retract at the plane of amputation [61,74,75], and a wound epithelium is quickly established over the site of injury [62,74]. Work by Zhang et al. 2015 suggests that in zebrafish jaw regeneration foxi1-foxo1b-pou3f1 signaling promotes wound epithelium formation. *Foxo1* is also upregulated in the early (24 h) stages of axolotl limb regeneration and appears to also be linked to the wound healing response in this organism as well [76].

This initial process of tissue degradation and wound epithelium formation is akin to that documented during limb regeneration in Urodele amphibians (as reviewed in [77]; however, unlike limb regeneration, no prominent innervation of the wound epithelium is apparent [61,62,64]), or required [63] for the continuation of the regenerative process. During limb regeneration, nerve-derived signaling in the wound site plays an essential role in blastema formation and growth; and nerve removal results in regenerative failure [78,79]. (reviewed in [80]). However, denervation of the already poorly innervated lower jaw of the adult newt has no effect on the regenerative process [63]. It is possible that direct nerve signaling is not necessarily required for regeneration, since salamander limbs that developed aneurogenically demonstrate regenerative abilities in response to amputation [69]. To explain the observations in these situations, the local tissues may compensate for low or absent innervation by the local production of sufficient regeneration-specific tropic factors [63]. More work is required to determine the necessity, nature, quantity and source of the trophic factors required for blastema formation in the context of jaw regeneration.

Early stages of limb regeneration in the newt and axolotl are relatively avascular, with the apical epithelial cap and initial blastema formation occurring in the absence of vascularization. As the blastema grows new vessels begin to invade from the proximal end, thus only later in blastema development is the entire structure fully vascularized [81]. Despite this delay in vascularization, one day post amputation transcriptional and gene ontology analysis indicates enrichment of genes relating to vasculature development in the regenerating axolotl [82]. This rapid vascular response in gene expression is also documented in the regenerating jaw, with transcriptional analyses suggesting that by two days post amputation blastemal angiogenesis and vasculogenesis is regulated by a pou5f1-cdx4-kdrl signaling pathway [83]. However, unlike the regenerating limb; new blood vessels sprouting from muscle tissue at the site of injury can be noted as early as 2 h post jaw amputation [83].

During limb regeneration in the salamander, the blastema is derived from a variety of stump cells [84,85,86]; although, not all stump tissues contribute equally to this progenitor population. For example, cells of connective tissue origin contribute more substantially to the blastema than muscle cells, while skeletal elements (bone and cartilage) fail to contribute at all [66,86,87]. Additionally, the contribution of these progenitor cells to the ultimate regenerate is restricted in a lineage specific manner. For example, muscle and Schwann cell derived blastema cells only differentiate into muscle and Schwann cells respectively, while blastema cells of connective tissue origin exhibit great plasticity and differentiate into cartilage, connective tissue and tendons [84,86,88].

A thorough understanding of the origins and differentiation potential of jaw blastema cells is largely lacking; however, it would appear from histological analysis, limited cell linage identification, transcriptional analysis and BrdU studies, that similar to limb regeneration, a heterogeneous population of dedifferentiated cells is generated at the plane of amputation in the jaw. These cells are derived from a variety of mature stump tissues, including muscle [64], a portion of which are *Pax7* positive muscle progenitor cells in newt [59] and *Pax3a, Isl1* muscle positive progenitor cells in fish [62], periosteum [89], and Foxi1 positive neural crest derived cells [62]. The *Pax3a, Isl1* positive progenitor population in fish localize to the posterior region of the regenerating jaw and are believed to contribute to muscle regeneration [62]. The *Foxi1* progenitors, associated with *Sox9a* expression 5 days post amputation, are localized to the region of new cartilage formation, suggesting that these cells contribute to skeletal regeneration [62]. Additionally, about 50% of the newt jaw blastema was found to exhibit lineage marks for dental pulp and dermal fibroblasts markers [61]. 

Cells of connective tissue origin also contribute to the jaw blastema. During limb regeneration, the missing skeletal tissues are derived completely from cells of connective tissue origin except for bone and cartilage [86,89]. In contrast, it appears that a small number of cells within the cartilage of the mandible undergo proliferation in response to amputation and may therefore contribute to the jaw blastema [59]. Cell lineage tracking experiments are required to determine the identity of these cells, to ensure they are not of soft connective tissue origin, and then test whether they migrate into the growing blastema. However, if the mandibular cartilage does contribute to the blastema it may indicate that different skeletal populations have different propensities for contributing towards a regenerative response. Secondly, from the available data, jaw regeneration in the newt appears dependent on the presence of the mandible. Excision of the central portion of the lower jaw, such that the mandible is left intact, results in regeneration, whereas an amputation that removes the entire mandible inhibits regeneration completely [90].

It is possible that different regions of the mandible contribute in different ways to the regenerating jaw. Histological analysis reveals that only peripheral tissues that are attached to the inner edges of the mandibles contributed towards the growth of the blastema, while little to no direct cellular contribution comes from posterior soft tissues [68]. Additionally, connective tissues cells play multiple roles during epimorphic regeneration. During limb regeneration, connective tissue cells not only contribute to all of the various connective tissues in the limb (dermis, tendons, periosteum, cartilage, bone) but they also play an essential role in the patterning of the blastemal cells into the missing limb structures, and provide positional cues to the surrounding cells to integrate the regenerated tissues [86] (reviewed in [77]). In newts, amputations have been performed along the length of the jaw; as the site of amputation approaches the proximal end of the jaw regenerative capacity diminishes, along with pattern [64]. Since the periosteum of the mandible has been shown to contribute to the regenerating jaw [74,89], it is therefore plausible that the periosteum of the mandible is necessary for pattern formation in jaw regeneration as well.

In summary, jaw regeneration requires the contribution of multiple cell types, derived from the underlying stump tissues, to the blastema. Mandibular tissue appears to be specifically required for the patterning and regeneration of the missing mandibular skeleton. We will further discuss the implications of these observations and how they can apply to regenerative therapies for humans below.

## 9. Jaw Blastema Cell Differentiation

### 9.1. Skeleton Formation

As the newt jaw blastema grows, the first differentiated cell type to arise is the chondrocyte. These cells produce a dense collagen matrix to establish a cartilaginous mass at the inner margins of the pre-articular bone of the stump, which is not continuous with the dentary bone or the Meckle’s cartilage [64,89]. As the cartilage grows outwards in a mediodistal direction, it fuses at the midline in a curved fashion to restore the general shape of the jaw [64]. At this later stage of regeneration in the adult and juvenile newt, the new cartilage finally becomes continuous with the Meckel’s cartilage, [64]. The cartilage bridge that ultimately spans the jaw, acts as a scaffold for mandibular bone formation. However, bone formation is distinctly different between fish and newts during jaw regeneration.

In zebrafish the entire cartilage scaffold is ossified through the process intramembranous ossification. No Meckle’s cartilage persists, and the median symphysis is absent due to the two mandibular bones fusing directly [62,89]. The remaining un-amputated portion of jaw however still retains cartilage, suggesting that any signals or stimuli responsible for the aberrant ossification pattern documented during jaw regeneration, relative to the original jaw, is restricted spatially to the regenerated tissue alone. At a molecular level, cartilage formation and subsequent ossification during regeneration in the zebrafish is, at least in part, dependent on Indian hedge homology a (*Ihha*) [89]. This is particularly interesting as *Ihha* is dispensable for zebrafish jaw development, where the only obvious defect in *Ihha* mutants was a delay in perichondral bone formation [89]. Therefore, the involvement and role of specific molecular signals involved in jaw skeleton regeneration may, in part, be different relative to their developmental counterparts. 

Age and/or metamorphosis appear to alter the regenerative capacity during newt jaw regeneration. In the juvenile newt, the cartilaginous mandible has both dentary and prearticular bone form around it, such that it resembles the original jaw exactly. After the ossification of the dentary bone a complete row of monocuspid teeth develop [61]. In the post-metamorphic adult newt, a thin layer of bone is formed along the ventral and lateral side in a proximo-distal direction of the regenerated cartilage scaffold, through intramembranous ossification, which fuses at the midline. Once dental lamina restoration is complete, bicuspid tooth development is observed in the newly formed dentary bone [61]. Intriguingly, no ossification occurs on the lingual side of the jaw [61]. Additionally, neither the pre-articular bone nor lingual dentary bone appear to regenerate [61]. Similarly, regenerates of the upper jaw which were missing parts of the nasal bone and the vomer are replaced with cartilage and not bone [61]. The molecular basis for this unusual ossification pattern, relative to the original jaw, is unknown; although, it is plausible that ossification of the outstanding cartilaginous skeletal structures may occur at some later stage. Together this data indicates that although newts retain impressive regenerative abilities as adults, the regeneration of a complete jaw structure occurs only in younger animals. 

Last, there are clear limitations in the regenerative and restoration capabilities of different skeletal elements within the jaw. In addition to the mandibular skeleton, the lower jaw of the fish and amphibians is also constituted by the more proximally located branchial and hyoid skeletons. Studies involving the removal of portions of the branchial and hyoid skeletal elements are limited to adult newts and indicate that these structures cannot be regenerated [64,78]. In the axolotl, cartilage formation in response to a tissue puncture through the hyoid apparatus in the tongue is also incomplete [91]. To our knowledge, the regenerative capacities of the branchial and hyoid skeletons have not been considered in the zebrafish or juvenile Urodeles. Thus, whether regeneration of these elements is related to organism age or is species specific remains to be resolved experimentally.

### 9.2. Teeth Restoration

Tooth development begins around the time of mandible bone formation [61]; however, the dental lamina required for this process is not generated *de novo* during regeneration. Instead, stump dental lamina migrates into the regenerating tissue to repopulate the new jaw [74]. If the stump is devoid of dental lamina, an edentulous jaw is generated [74]. Quarter jaw amputations have revealed that immigration of dental lamina cells into the adult regenerate is directional, only occurring from the posterior stump towards the anterior of the regenerate [74]. This directional mechanism is age specific, as identical amputations in juvenile newt species are associated with migration of dental lamina cell into the regenerate from both the anterior and posterior stump tissue [74]. The mechanisms controlling and restricting this directional migration of dental lamina is unknown. An important note to make is that newt dentation is restored in an age appropriate manner during regeneration, with the adult jaw regenerating bicuspid and the juvenile jaw monocupsid teeth. It is thought that the morphology of the dentation is hormonally regulated [61].

## 10. Pattern Formation and Positional Identity

Although the regenerated jaws in the zebrafish and adult newt are functionally restored, aesthetically they are not. The general, curved shape of the jaw is reformed in both model organisms, with dentation when appropriate, but the dimensions, precise pattern and proportions of the regenerate are frequently distorted relative to the original [64,89]. Micro computed tomography (µCT) scans of these regenerated jaws demonstrate an irregular and rough pattern, unlike that of the original [59,89]. µCT scans have not been used to assess the regenerate of larval newts; however, based on whole mount staining the regenerate matches the original mandible identically [61]. The reason for this patterning difference between larval and adult newts is unknown, but may be a consequence of metamorphosis-related cellular alterations or differences in mandible structure and tissue composition at these different life stages. Identification of the mechanisms responsible for this age-related patterning defect, and subsequent manipulation thereof, will be of great value in the context of human regenerative medicine.

### Hox Gene Re-Expression and Positional Plasticity

There is great value associated with understanding the transcriptional profile of blastema cells, and the regulatory mechanisms thereof, as these cells re-pattern, re-differentiate and integrate. The spatiotemporal differential expression patterns have been documented for a plethora of genes in this context, including those involved in wound healing and development [80,82,92,93,94]. These studies implicate multiple signaling pathways and transcription factors during pattern formation in the regenerate. Here we will focus on patterning genes, including the *Hox* genes, because they have been well documented to play an important role in both pattern formation and the retention of positional memory in mature adult cells [95,96].

The *Hox* transcription factors are expressed sequentially along the length of the body axis and participate in body patterning and the establishment of positional identity during embryogenesis [97]. The concept of positional identity is well described in this context; however, a paucity of understanding on this topic exists in adult tissues. Available data indicates that mammalian and amphibian cells of connective tissue origin retain positional information throughout adulthood [86,88,95,98,99]. Adult human fibroblasts, being the most-well characterized cell type in this context, stably retain the expression of specific *Hox* genes, even if placed in cell culture, in a pattern similar to that document in the developing embryo [95,98,100]. A variety of epigenetic mechanisms control this phenomenon [96].

As we have described previously, connective tissue cells are essential for the patterning of the missing structures. Moreover, these cells provide positional cues to the surrounding tissues during both embryogenesis and regeneration ([101] and reviewed in [77]). However, little information is known about how connective tissue cells communicate positional formation to each other or other cells types. In humans, dermal fibroblasts communicated positional information to epidermal cells through the secretion of Wnt molecules [100]. Additionally, some data indicates that heparin sulfate proteoglycans communicate positional information between connective tissue cells in the axolotl [86,102]. Thus, more research is required to understand how cells with positional memory use this memory to pattern the regenerating structures. 

While connective tissue cells must retain positional memory during homeostasis, these cells must also exhibit the capacity of positional re-programing, such that they can adopt new positional information as they contribute to more distant structures in the regenerate. In support of this hypothesis, positional plasticity has been documented during axolotl limb regeneration. By grafting differently staged blastemas to different locations on the limb it was observed that early blastemas patterned and expressed molecular markers consistent with the new location. Late staged blastemas, however, retained memory of their original location and patterned according to the site of origin. These observations indicate that early stage blastema tissue is positionally plastic, while late staged blastema tissue, that is beginning to undergo differentiation, has a new positional identity already established [103].

Furthermore, plasticity in tissue polarity has also been exhibited during both jaw and limb regeneration. If salamander the autopod (hand) is attached to the salamander flank, and the limb is amputated, the remaining stump tissue has a reverse proximal/distal polarity. The resulting limb regenerate does not have a reversed polarity however, it regenerates with a normal proximal distal axis, indicating that proximal distal polarity in the limb is plastic (Figure 2) [104]. A similar process occurs in the adult newt jaw. When a quarter jaw fragment is flipped, relative to the host site, and transplanted from the left to the right mandible and amputated, the newly regenerated jaw will have the appropriate polarity based on the new host location [74]. Moreover, the dental lamina, which normally migrates into the regenerate in only the posterior to anterior direction, will also migrate according to the new host location (Figure 3). This suggests that the polarity of the dental lamina has also been reprogrammed at the site of amputation [74]. The mechanisms for this reprogramming event are, however, unknown but it can be postulated to involve changes at an epigenetic level.

As part of this re-programing event, adult cells would need to assume location-appropriate *Hox* expression profiles; and various studies have demonstrated dramatic changes in transcriptional profile of *Hox* genes during regeneration (Figure 4). In the regenerating axolotl (*Ambystoma mexicanum*) and larval newt (*Notophthalmus viridescens*) limb, Hox genes involved in normal patterning of the limb bud during development—*Hoxa9, Hox13, Hoxb13, Hoxc6, Hoxd8, Hoxd10,* and *Hoxd11*—were re-expressed in the blastema [105,106,107]. In the zebrafish, several caudal fin specific *Hox* genes, including *dlx4a* and *dlx5a*, were expressed in the blastema during the regeneration of this structure as well [94,108]. In the case of the zebrafish, molecular analysis correlates this re-expression pattern with a loss in H3K27me3 marks near the start site of these genes [94].

Alterations in *Hox* gene expression have also been documented in the regenerating mandible. In response to amputation of the distal third of the lower jaw in zebrafish, 22 of the annotated Hox genes were also found to be differentially activated [62,83]. As early as 2 h post amputation the expression of *Hoxb2a, Hoxb5a, Hoxa9a* and *Hoxb13a* was elevated; with the remaining genes upregulated by day 5. This expression pattern is somewhat surprising considering that the mandibular arch is *Hox* negative during early embryogenesis. However, it was also observed that *dlx1*, a homeobox gene specifically expressed in the mandibular arch of the CNC, expression is increased in the fish jaw blastema. RNA in situ hybridization indicates that *Hoxa2b* was expressed in the blastema mesenchyme, localized most strongly in the chondrogenic zone [62]. Additionally, 5’ *Hox* genes (including *Hoxa4a, Hoxa5a, Hoxa9a, Hoxa9b, Hoxb13a, Hoxa11b* and *Hoxd11a*), associated with patterning of posterior structures, were also expressed [62]. Fish hematopoietic progenitors express these 5’ *Hox* genes; thus, invading blood cells may in part explain the elevated expression of these genes within the blastema. However, RNA in situ hybridization indicates that *Hox11* was broadly expressed in the blastema mesenchyme, concentrated near the wound epithelium as well as adjacent the stump muscle [62]. This activation of positionally-unrelated *Hox* genes has been documented during limb and tail regeneration as well [94,106] and may be a consequence of the positional reprogramming that is required for re-patterning to occur.

The capacity of cells to assume new positional information is largely limited to a regeneration permissive environment. However, in the context of stem cell grafts, *Hox*-negative cells can assume new identities while *Hox*-positive populations cannot (Figure 5). In the developing murine limb bud, *Hoxa11* is expressed in the mesenchyme, and this expression is retained in the adult murine tibia, specifically in periosteum and osteocytes, even in response to injury and subsequent repair. This is unlike the mandible, where the skeleton retains a *Hox11a*-negative status after development, even in response to injury. When *Hox11a* positive tibia derived periosteum was grafted into the *Hox11a*-negative mandibular environment, *Hox11a* expression was retained in the grafted population. These cells failed to differentiate into osteoblasts and thus did not integrate with the host tissue. Instead a cartilaginous callus was generated as the junction site [109]. The reciprocal grafting experiment, however, gave a very different outcome. *Hox11a*-negative mandible derived periosteum grafted into the *Hox*-positive tibia resulted in appropriate intramembranous bone formation, with the grafted cells expressing *Hoxa11* and integrating with the surrounding host tissue seamlessly [109]. These observations indicate that the Hox11a-negative (mandibular) population is more positionally plastic than the *Hox11a* positive (tibia) population, and that this plasticity promotes the integration of the transplantation into the host site. However more studies are required to test this hypothesis. For example, a complete *Hox* expression analysis should be performed to determine whether the up-regulation of other *Hox* genes occurs in the injured mouse mandible, in a similar manner as the zebrafish. It would also be very interesting to observe changes in the epigenetic landscape of the mouse mandibular and tibia transplants before and after transplantation to determine whether the tibia transplants are resistant to epigenetic alterations.

There are, however, limitations to this reprogramming response; which apparently prevents tissues from assuming positional identities of distant, unrelated body locations. For example, in the axolotl, when head-derived full thickness skin or dermis alone was grafted into the limb and the limb was then amputated at the site of transplantation, regeneration was inhibited or impeded respectively [66]. Head derived cells did contribute to the blastema; however, in the case of regeneration failure the structure never developed beyond the early bud stage [66].

Altogether, this data indicates that achieving plasticity of positional memory plays an important role in the regeneration of the jaw. It is likely that future regenerative therapies will need to recapitulate this process to promote endogenous regeneration in the jaw. 

## 11. Mechanisms of Jaw Fracture Healing

As opposed to endogenous regeneration of the jaw, the healing of fractured jawbones occurs poorly in both mammalian and regenerative model organisms (Table 1). Multiple studies in rodent and avian models indicate that skull and jaw fractures heal using secondary ossification where the fractured ends of the bones become connected through a fibrous callus [110,111,112]. Cartilage and bone formation at the fractured bone ends is documented in the chick, generated by periosteal cells as well as bone derived osteoblasts or osteocyte [112]. In the rodent, cartilage formation is absent but bone formation is achieved by proliferating parietal bone-derived osteoblasts that invade the callus; however, this invasion is also restricted to sites adjacent the damaged bone ends [110,111]. This limited invasion of skeleton-derived cells is thought be due to the fibrotic nature of callus [111]. Interestingly, increased inflammation correlates to increased bone formation during fracture repair in the rodent [111]. Since the callus fails to align the fractured ends of the bones, the integration of the new and old tissue is poor and prevents the restoration of the original pattern [110,111,112].

While regenerative models do not heal bone fractures perfectly, their regenerative response is more robust than mammalian or avian species. In *Ambystoma maculatum*, periosteal cells respond to the initial injury, covering the fractured ends of the bone and ultimately joining the dentary and prearticular sides of the fracture with a bony bridge. A cartilage callus is generated separately to this bridge, and appears to be derived by metaplasia of the surrounding connective tissue rather than being derived from the Meckle’s cartilage or bone directly. Over time, the callus grows to unite the two fractured portions of the jawbone and ossifies. Although this healing response is more robust relative to that of mammals and birds, this structure is not well integrated with the original jaw. The bony barrier that divides the callus cartilage and Meckle’s cartilage is retained and jaw misalignment is frequently documented [113]. Similarly, acellular and cellular fish skeletons responds similarly to injury; and lower jaw fractions are resolved by the formation of a cartilaginous callus with limited ossification [114]. Interestingly, the efficiency of callus formation is species dependent and strongly affected by diet, water quality and temperature [114].

## 12. Tissue Integration

Tissue integration is an essential but under-appreciated topic in context of regenerative medicine. Bone as well as other jaw tissues, including muscle and nerves, need to integrate seamlessly with one another during regeneration or after a tissue engraftment, to restore functionality of the resulting structure. For example, the regeneration of missing portions of the mandible that effectively restores the shape of the jaw but fails to reconnect appropriately with the stump tissue, would impede, if not hinder, its use in mastication and speech. 

Tissue integration and regeneration are often considered synonymous events; however, the two process can be uncoupled (as reviewed in [115]). This observation suggests that distinct molecular processes must underlay each phenomenon individually. Additional support for this idea comes from situations where regenerative responses at different locations, in the same animal, exhibit differential capacities for integration. For example, limb regeneration in the Urodele naturally exhibits full integration of the regenerate with the stump tissue (as reviewed by [115]), while the adult jaw does not. The initial cartilaginous structure regenerated within the post-metamorphic newt jaw does become continuous with the Meckel’s cartilage; but no pre-articular bone nor lingual dentary bone are generated [61,64], thus preventing appropriate connections between the stump pre-articular and lingual dentary bones to the regenerate. Interestingly, pre-metamorphic newts not only regenerate all of the missing jaw structure, but they also appear to be integrated perfectly with the stump structures [61]. Thus, metamorphosis and/or aging induces changes in the cells that inhibits integration. Understanding these changes will help us understand how we can promote better integration of new jaw structures in mammals. 

Jaw fracture repairs in the newt exhibits even worse integration, relative to jaw regeneration, with the misaligned bony union at the site of injury being separated by a bony barrier from the jaw bone [113]. Integration failure in this context appears to be conserved, with mammalian fractures also exhibiting callus formation that does not restore appropriate pattern or tissue connections [110,111,112]. Despite the importance of integration to regeneration, the molecular pathways and mechanisms for this phenomenon still need to be elucidated. 

## 13. On the Development of New Regenerative Therapies: Conserved Mechanisms of Endogenous Regeneration

### 13.1. Positional Plasticity

Ideally, the repair of damaged or missing patient jaw tissue would occur through the induction of an endogenous regenerative response. Endogenous regeneration alleviates the need to harvest tissue from the patient or a donor, and avoids additional issues such as the rejection of the tissue transplant. Furthermore, in regenerative models, endogenous regenerative responses result in the formation of a structure that is similar in pattern to the original structure, and is better integrated with the original tissues including the surrounding muscles, nerves, and vasculature. Here we have reviewed how jaw regeneration occurs in multiple regenerative species, and have compared these mechanisms with amphibian limb regeneration to help identify unified regenerative mechanisms. 

One of the most important unifying properties of regenerating limbs and jaws appears to be the plasticity of positional information. Evidence indicates that plasticity not only plays an important role in establishing the pattern of the missing structures, but also in the integration of the new tissue with the existing tissue. Positional plasticity is achieved naturally in a regenerating environment, but little is known about how it is established. In regenerating limbs, there is evidence that nerve-derived signals help maintain positional plasticity in blastema cells [80,93,103], and that cells that are differentiated or are differentiating are not positionally plastic [103,116]. Both the jaw and limb blastemas appear to re-express patterning genes, including patterning genes that are not expressed in these tissues during embryogenesis, although more experimentation will be required to understand the full extent of these expression patterns in multiple model systems. This reactivation of patterning genes is likely due, in part, to large scale opening up of the chromatin landscape in the blastemal cells. Although this has not been looked at in detail in the jaw, increased presence of euchromatin has been well-established in the limb regenerate [117,118]. Additionally, multiple epigenetic modifications have been observed in different species and regenerating structures, and appear to play a role in regenerative ability. 

How can positionally plastic human cells be generated? In tissue culture, tissue plasticity can be induced in adult mammalian fibroblasts through the transfection of specific combinations of pluripotent transcription factors, which converts the cells into induced Pluripotent Stem Cells (iPSCs) [119]. Interestingly, however, Hox genes are differentially expressed in iPSCs that have been generated from different somatic sources, indicating that these cells retain some positional memory [65,67]. One possibility is to generate iPSCs from a source of cells that are known to be positionally plastic. For example, in mice, transplants of mandibular tissue adopt the new Hox code of the host site, indicating that these cells are inherently plastic. This is exciting, because it poses the jaw as a structure that could be uniquely well suited for the generation of positionally plastic jaw progenitor cells in vivo. 

To date, multiple cell types have been converted to iPSCs in vivo [120,121,122,123,124,125,126]*.* In vivo reprograming of mesenchymal cells into iPSCs that differentiate into epithelial tissue has been used to heal skin ulcers in mice [126]. The use of this technology has also been applied with some success in regenerating missing neurons in damaged adult brains, although the efficacy of reprogramming and the survival of the differentiated neurons depends largely on the in vivo environment and type of injury [122]. However, concerns of the use of iPSCs and their tumorigenic potential need to be resolved before this technology can be safely used for in vivo regeneration in humans. Additionally, it should be noted that during endogenous regeneration, cells are reprogramed to a progenitor-like state as opposed to a pluripotent embryonic-like state, yet are able to establish positional plasticity. Thus, there appears to be unique mechanisms in the erasure of tissue identity versus positional identity to make blastema cells plastic. Further research focused on understanding how blastema cells attain positional plasticity while remaining restricted in their potential lineages will provide valuable insight into the development of endogenous regeneration technologies. 

### 13.2. Imperfect Fracture Repair

One surprising observation is that both regenerative models and poorly regenerating species, including mammals, do not heal jaw fractures well. Although regenerative species heal fractures better than humans, the integration of the fractured ends of the jaw is not perfect. In these species it appears that larger injuries that require an endogenous regenerative response do a better job of seamlessly connecting the new tissue with the existing tissue. This could be due, in part, due to the cells that help heal jaw fractures versus those that contribute to a regenerative response. The cells that contribute to jaw fracture repair appear to be derived mostly from the periosteum and bones. Jaw regeneration however recruits multiple cell types, including different connective tissues, muscle progenitors, cranial neural crest derived cells, and dental pulp cells. It is possible that a more diverse contribution is required for improved integration. 

Another potentially important aspect is that *de novo* pattern formation occurs during endogenous regenerative responses, but is unlikely to occur during jaw fracture repair. No blastema forms during fracture repair, and it seems that this response is more attune to a compensatory regenerative response as opposed to epimorphic regeneration. However, a more detailed analysis of *Hox* gene expression patterns needs to be performed during jaw fracture healing in different species to fully understand how pattern information might be modulated during fracture repair. Regardless, the signaling events that occur during pattern formation are unlikely to be present in fractures, and this could also explain why fractures integrate poorly. 

### 13.3. Age-Related Changes in Regenerative Capacity

Another unifying property associated with better regenerative outcomes is young age. This has been well documented with human patients that have received reconstructive jaw surgeries, where the new tissue is well integrated, and sometimes appears to stimulate an endogenous regenerative response. It is also observed in vertebrate regenerative models such as the newt, where juveniles replace and integrate the missing jaw tissue perfectly, while adult newts only replace some of the missing jaw structures. It has been previously hypothesized that the presence of a sufficient supply of bone precursor cells is a major contributor to the improved outcomes in younger patients. However, as we have learned from organisms that regenerate endogenously, the establishment of cellular plasticity as well as the recruitment of a more diverse cell population during a regenerative response may also be contributing factors. 

### 13.4. Transplantations

While being able to endogenously regenerate damaged or missing jaw tissue would be ideal, there is still a lot we need to learn about the basic biology underlying these mechanisms before we can apply this to human medicine. Bone transplantation methodologies have been well developed over the years, and have a relatively high success rate. Due to the large number of bone grafting procedures conducted (estimated to be more than 500,000 in America annually) (AAOS, 2010), there is sufficient evidence to support idea that an effective bone graft should not only fill in the deficit but also promote new bone formation on its surface (osteoconduction) and within itself (osteogensis); stimulate the adjacent, host tissue stem cell population to generate new bone cells (osteoinduction), and integrate with the host bone without any intervening fibrotic layer [127,128]. Of autologous, allogenic and synthetic bone grafts, only the former fulfills all these requirements and is therefore considered the “gold standard”, in addition to being histocompatible (as reviewed in [129]). 

One feature that appears to play a role in the efficacy of an autologous transplantation is the location from which the grafted tissue was harvested. Evidence indicates that bone harvested from the head (cavarial and mandible) exhibits better integration and stimulates more bone growth when transplanted into the jaw compared to bones harvested from more distant regions in the body (iliac or tibia). It is possible that this is due to the increased positional plasticity of head bone cells, as we have described previously, compared to bones from other regions. It may also be related to how these bones naturally form during embryogenesis. The mandible and cavarial bones develop via intramembranous ossification, while the iliac and long bones develop via endochondrial ossification. While callus formation appears to be a conserved mechanism of bone repair regardless of location, physiological differences in these bones, and which cells contribute to bone repair may play a role in how compatible they will be as a transplant for jaw repair.

Along with tissue viability, the healing and integration of the transplanted bone with the existing bone is potentially the greatest challenge to a successful outcome. While increased viability will depend on the health of the transplantation tissue and the host site, it is also likely that integration will play a role because the vasculature and neural connections in the host location will need to integrate with the engrafted tissue for its survival. One potential way to facilitate this process is by combining the induction of an endogenous regenerative response in the host location by inducing pluripotency in vivo along with an autologous transplant from the head skeleton. In theory, this would deconstruct the surrounding tissues and make the cells more plastic, and thus more likely to integrate with the bone transplant. Testing this method in regenerative and non-regenerative model organisms will help us determine whether this is a feasible method for jaw-related regenerative therapies.

## 14. Conclusions

Unlike humans, multiple vertebrate organisms, including the zebrafish, newt, and axolotl, naturally exhibit profound capabilities in terms of regenerating injured or missing jaw tissue. Utilizing these model systems, we can begin to understand as well as identify conserved mechanisms underlying regeneration, including, but not limited to, the mechanisms of dedifferentiation, positional plasticity, *de novo* pattern formation, and tissue integration. Translating this knowledge into the mammalian system will allow us to engineer better regenerative therapies for humans suffering from a variety of jaw aliments, and in some cases it already has. Future research is additionally required to thoroughly compare and contrast the mechanisms and outcomes of jaw injury and transplantation in regenerative and non-regenerative models so to determine and understand the pathways that can potentially be deployed to facilitate a more robust regenerative response in non-regenerative species.

## Figures and Tables

**Figure 1 ijms-19-03752-f001:**
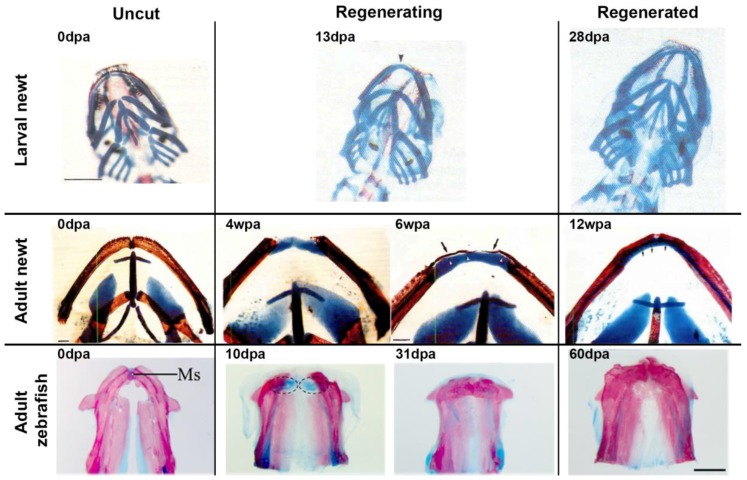
Whole mount preparations demonstrate the jaw-regenerative capacities of Urodeles and Zebrafish. Stage 40 newt larvae (top panel) exhibit more robust regenerative capabilities relative to their adult counter parts (middle panel) as well as to zebrafish (lower panel). In all three cases a cartilage rod is initially generated, fusing the two damaged ends of the mandible; however, this cartilage is replaced completely by bone in the zebrafish and results in the loss of the mandibular symphysis (MS). In the larval newt the regenerated pattern is identical to the original jaw; while in the adult regenerate is inappropriately patterned in terms of bone and cartilage relative to the original. Blue—cartilage staining, red—bone staining. Arrow in upper panel highlights the absence of dentary bone 13 dpa. Arrows in the middle panel highlights the dentary bone growing towards the mid-line 6wpa, and the absence of ossification on the lingual side of the jaw at 12 wpa. Dpa—days post amputation, wpa—weeks post amputation. (Images modified with permission from Ghosh et al. 1994 and Wang et al. 2012).

**Figure 2 ijms-19-03752-f002:**
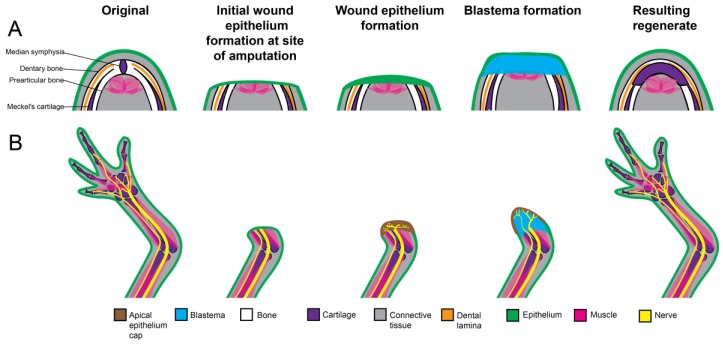
Differences and similarities in blastema and structure formation during adult Urodele jaw and limb regeneration. (**A**) The stages of jaw regeneration. The intact amphibian jaw consists of Meckel’s cartilage, dentary and prearticular bone, and the median symphysis connecting both sides of the jaw. The hyoid and brachial skeletons have been removed for simplicity. Upon amputation, a wound epithelium forms over the remaining jaw stump tissue. Independent of nerve singling, blastema cells accumulate at the severed edge of the jaw, derived from a variety of mature stump tissues, and will proliferate, repattern, and differentiate into a structure similar to the original structure, but with notable differences. The medium symphysis is lost and the majority of regenerated skeleton is cartilaginous (connect with Meckel’s cartilage), with only the dentary bone being restored. (**B**) The stages of limb regeneration. Limb regeneration also requires blastema formation, however its induction is slightly different from that of jaw regeneration. After amputation, a wound epithelium (WE) forms over the severed edge of the limb stump. The WE becomes innervated by the regenerating neurons, and establishes a specialized wound epithelium known as the Apical Epithelial Cap (AEC). The AEC is required for the dedifferentiation of the limb stump cells into blastema cells. The blastema cells, also derived from a variety of mature tissue, proliferate, pattern, and redifferentiate into all of the missing limb structures. The resulting limb regenerate is identical to the original.

**Figure 3 ijms-19-03752-f003:**
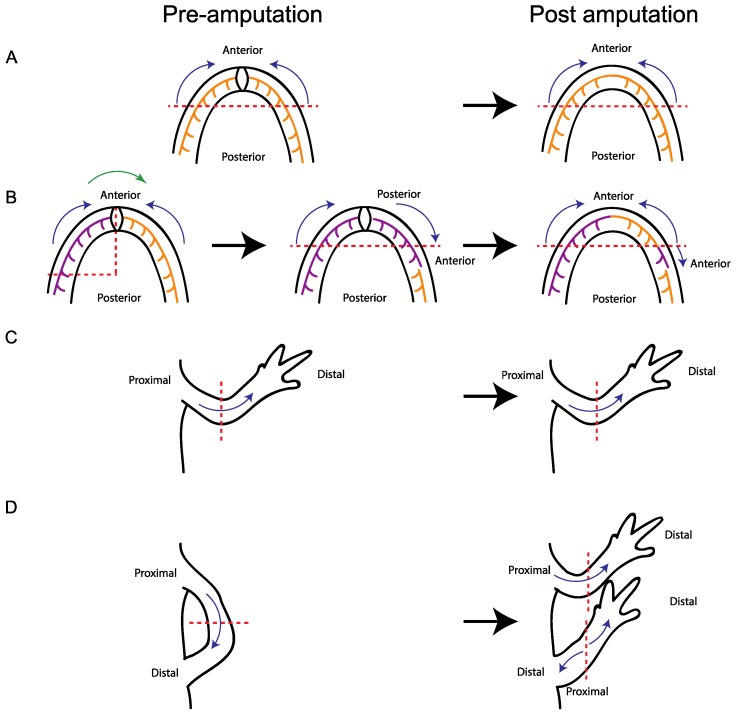
Plasticity of tissue polarity during jaw and limb regeneration is conserved. (**A**) Jaw regeneration normally occurs with a posterior to anterior polarity; (**B**) When a portion of the jaw is reversed along its anterior-posterior axis and amputated, rather than generating structures that are posterior to the amputation plane, the blastema cells reverse their polarity so that they only regenerate structures anterior to the amputation plane. Dental lamina migrates into the adult jaw regenerate in a unidirectional manner, from posterior to anterior; however, the dental lamina migrates into the regenerate in the appropriate direction, despite the grafted tissue being rotated relative to the host site and regenerate; (**C**) During normal limb regeneration, the regenerate retains the proximal/distal polarity of the original limb; (**D**) Limbs with reversed proximal/distal axes can be generated by attaching the autopod to the flank. When the limb is amputated, the part of the limb that remains attached to the flank has the reverse proximal/distal orientation. However, rather than regenerating limb structures proximal to the wound site, the blastema reverses its polarity such that it regenerates tissue distal to the amputation plane. Therefore, there is a conserved mechanism, which prevents regeneration in a proximal direction. Red dashed lines represent the plane of amputation. Blue arrows indicate the direction of tissue polarity. Purple and orange lines represent dental lamina. Green arrow represents tissue grafting. Black arrows represent amputation and regeneration time.

**Figure 4 ijms-19-03752-f004:**
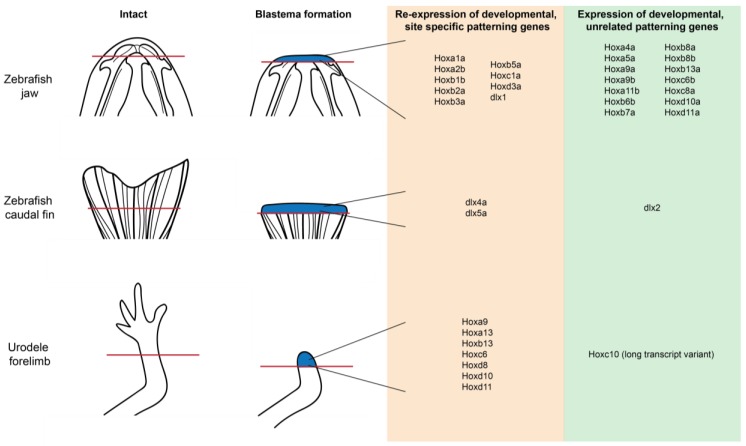
Hox gene re-expression is a characteristic of regeneration shared between species and anatomical locations. Jaw, limb and tail blastema tissue exhibit the re-expression of both developmentally related and unrelated Hox genes. Red line indicates plan of amputation; blue tissue indicates blastema tissue.

**Figure 5 ijms-19-03752-f005:**
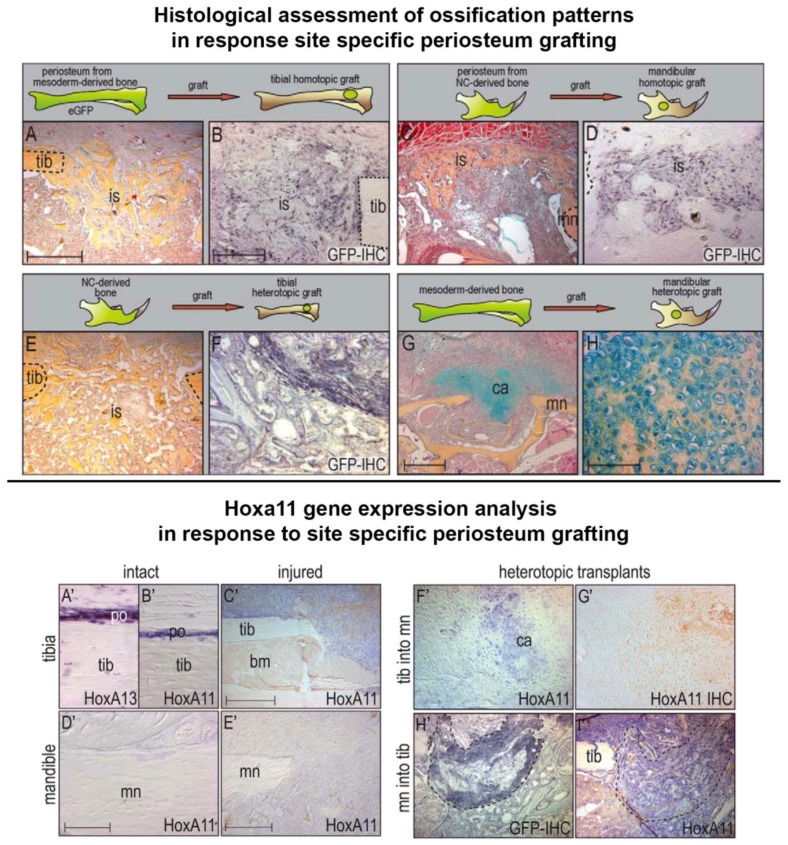
The capacity of bone derived mesenchymal cells to participate in bone restoration is dependent on both the donor and recipient site. Top panel: Robust tibia bone repair, whereby pentachrome staining indicates intramembranous ossification (**A**), is facilitated directly by the homographic translation, survival and differentiation of GFP-positive tibial periosteum at the site of injury (**B**, GFP immunohistochemistry). A similar response is documented in mandibular sites (**C**) transplanted with GFP-positive mandibular periosteum (**D**). Although intramembranous bone is generated in tibial sites (**E**) in response to engraftment, survival and differentiation of GFP positive mandibular derived periosteum cells (**F**); tibial periosteum transplanted into the mandible exhibits failed integration due to inappropriate endochondral ossification (**G**) and cartilage condensation (**H**, high magnification). ca, cartilage; is, injury site; mn, mandible; tib, tibia. Scale bar: 200 µm in A, C, E; 100 µm in B, D, F; 400 µm in G; 50 µm in H. Lower panel: Periosteum derived from the adult tibia expresses *Hoxa13* (**A’**) and *Hoxa11* (**B’**) under homeostatic conditions, and *Hoxa11* in response to injury (**C’**). *Hoxa11* expression is absent in mandibular periosteum under both normal and injury conditions (**D’**,**E’** respectively). Tibia periosteum retains *Hoxa11* expression when transplanted into the Hox-negative mandibular environment (**F’**,**G’**), while GFP-positive mandibular periosteum (**H’**, GFP expression encompassed by dashed lines) grafted into the tibia exhibits *Hoxa11* expression (I, mandibular cells encompassed by dashed lines) bm, bone marrow; ca, cartilage; mn, mandible; po, periosteum; tib, tibia. Scale bar: 50 µm in A, B, D, F, G; 100 µm in E; 200 µm in C, H, I. (Images republished with permission from Leucht et al. 2008).

**Table 1 ijms-19-03752-t001:** Comparison of regenerative and fracture healing responses across vertebrate species in the lower jaw. n/a—not applicable; w/m—whole mount staining; µCT—Micro computed tomography, hst—histology.

Process		Mammalian	Avian	Urodele	Teleost
**Regeneration**	Capacity	No	No	Yes	Yes
	Mechanism	n/a	n/a	Epimorphic regeneration	Epimorphic regeneration
	Pattern relative to original	n/a	n/a	Juvenile = identical (based on w/m), Adult = different (based on w/m, µCT)	Different (based on w/m)
	Integration with stump	n/a	n/a	Juvenile = fully integrated (based on w/m) Adult = incomplete (based on w/m, hst)	Incomplete (based on w/m, hst)
**Fracture healing**	Capacity	Yes	Yes	Yes	Yes
	Mechanism	Secondary ossification	Secondary ossification	Absence of secondary ossification	Absence of secondary ossification
	Pattern relative to original	Poor (based on hst, µCT)	Poor (based on hst)	Poor (based on hst)	Poor (based on hst)
	Integration with stump	Poor (based on hst, µCT)	Poor (based on hst)	Poor (based on hst)	Poor (based on hst)

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
