# Peer review of "Regenerative Models for the Integration and Regeneration of Head Skeletal Tissues"

_ijms, 2018, doi:10.3390/ijms19123752_

Round 1

Reviewer 1 Report

This is an excellent and comprehensive review of the biological aspects of craniofacial bone regeneration.  The manuscript is thorough and presents a detailed review of previous studies dealing with regional head skeletal tissue regeneration in multiple species, including mammals, fish, and amphibians. The manuscript also deals with cellular and molecular data as well. I liked the fact that the authors summarized some of the published data in Table format and also included a couple of mechanistic figures which collectively support key findings. Here are a few more comments to consider in order to strengthen the manuscript before it can be published in IJMS:

There are multiple      spelling errors (lines 125, 141, 171, 257, 692, 703, 704) and thus the document      needs editing.

I was surprised that the      authors did not include a section on alveolar bone regeneration as part of      their review.  Although, the      discussion on jaw regeneration was excellent, it would have been nice if      just one section was dedicated towards alveolar bone (since it is such an      integral part of the jaw; both mandible and maxilla).

A reference should be      given for sentences shown in lines 51-55.

n/a should be defined in      Table 1.

Line 555, Table should be      with a capital T.

The sentence in lines      267-268 should be revised as the word regeneration shows up 3 times.

Lines 270-271, it would be      good if the authors also provided the common name of the species.

Figure 1, I am surprised that      the authors included nerves but not blood vessels in their schematic.  As angiogenesis is such a critical      process of tissue regeneration, I encourage the authors to also include the      presence of blood vessels into their schematic.

9. I also encourage the authors to include some of the published data from various manuscripts that they cite so that the readers can visualize some of the processes.  For example, the description of the data on Hox expression (lines 507-522) would be enhanced if the authors include some images from the study they cite.  Again, this can be done for some of the other data that the authors describe as well. I believe that 3-4 data figures should suffice and will enhance the review.

Author Response

Reviewer 1: 

Thank you for your comments! 

1. There are multiple spelling errors (lines 125, 141, 171, 257, 692, 703, 704) and thus the document needs editing. A reference should be given for sentences shown in lines 51-55. Line 555, Table should be with a capital T.

We couldn’t find errors in all of the locations the reviewer identified –it appears that the line numbering was different from the version we and the reviewers had. However, we have more thoroughly edited the article and fixed all of the errors that we could find. 

2. I was surprised that the authors did not include a section on alveolar bone regeneration as part of their review.  Although, the discussion on jaw regeneration was excellent, it would have been nice if just one section was dedicated towards alveolar bone (since it is such an integral part of the jaw; both mandible and maxilla).

In order to address alveolar bone regeneration, text reviewing this topic was added to section 2 of the document.  

3. n/a should be defined in Table 1.

n/a has been defined in the legend of table 1.

4. The sentence in lines 267-268 should be revised as the word regeneration shows up 3 times.

The line was revised. 

5. Lines 270-271, it would be good if the authors also provided the common name of the species.

The common names were provided for the different species listed in Lines 270-271. 

6. Figure 1, I am surprised that the authors included nerves but not blood vessels in their schematic.  As angiogenesis is such a critical process of tissue regeneration, I encourage the authors to also include the presence of blood vessels into their schematic.

We tried to include blood vessel regeneration in the Figure 1; however, it ended up making the figure too complicated visually. Therefore, we decided to add additional text describing angiogenesis in the regenerate to the document. 

7. I also encourage the authors to include some of the published data from various manuscripts that they cite so that the readers can visualize some of the processes.  For example, the description of the data on Hox expression (lines 507-522) would be enhanced if the authors include some images from the study they cite.  Again, this can be done for some of the other data that the authors describe as well. I believe that 3-4 data figures should suffice and will enhance the review.

In order to address this concern we have added 3 additional figures. One is a graphical summary of the Hox expression data, to date, documented in blastema tissues from different anatomical locations. The other two images are taken from previous publications, we are awaiting permission from the respective authors for the use of this data.   

Reviewer 2 Report

Dear authors,

many thanks for the interesting publication.

In my opinion only minor revision is necessary as follows:

(1) Please check the content of paragraph 2.1. The therapeutic concepts for bone regeneration in the jaw are inadequate as no CaP-based alloplastic materials, allografts or xenografts are included. Please extend this paragraph and include these clinical concepts that are used in the daily practice.

(2) Please include a description of bone autografts, which include the hydroxyapatite-based matrix, the different bone cells and other related cell types such as osteomacs or endothelial cells and different proteins such as BMPs. 

Author Response

Reviewer 2: 

Thank you for your comments! 

8. (1) Please check the content of paragraph 2.1. The therapeutic concepts for bone regeneration in the jaw are inadequate as no CaP-based alloplastic materials, allografts or xenografts are included. Please extend this paragraph and include these clinical concepts that are used in the daily practice.

We have extended this paragraph as requested. 

9. (2) Please include a description of bone autografts, which include the hydroxyapatite-based matrix, the different bone cells and other related cell types such as osteomacs or endothelial cells and different proteins such as BMPs. 

We have included a description as requested.